# The gamma response to colour hue in humans: Evidence from MEG

**Gavin Perry** *, **Nathan W. Taylor**, **Philippa C. H. Bothwell**, **Colette C. Milbourn**, **Georgina Powell**, **Krish D. Singh**

Cardiff University Brain Research Imaging Centre (CUBRIC), School of Psychology, Cardiff University, Cardiff, United Kingdom

* perryg@cardiff.ac.uk

**Data Availability Statement:** Data in support of the paper is available at the following link: https://doi.org/10.6084/m9.figshare.13313276.v1.

## Abstract

It has recently been demonstrated through invasive electrophysiology that visual stimulation with extended patches of uniform colour generates pronounced gamma oscillations in the visual cortex of both macaques and humans. In this study we sought to discover if this oscillatory response to colour can be measured non-invasively in humans using magnetoencephalography. We were able to demonstrate increased gamma (40–70 Hz) power in response to full-screen stimulation with four different colour hues and found that the gamma response is particularly strong for long wavelength (i.e. red) stimulation, as was found in previous studies. However, we also found that gamma power in response to colour was generally weaker than the response to an identically sized luminance-defined grating. We also observed two additional responses in the gamma frequency: a lower frequency response around 25–35 Hz that showed fewer clear differences between conditions than the gamma response, and a higher frequency response around 70–100 Hz that was present for red stimulation but not for other colours. In a second experiment we sought to test whether differences in the gamma response between colour hues could be explained by their chromatic separation from the preceding display. We presented stimuli that alternated between each of the three pairings of the three primary colours (red, green, blue) at two levels of chromatic separation defined in the CIELUV colour space. We observed that the gamma response was significantly greater to high relative to low chromatic separation, but that at each level of separation the response was greater for both red-blue and red-green than for blue-green stimulation. Our findings suggest that the stronger gamma response to red stimulation cannot be wholly explained by the chromatic separation of the stimuli.

## Introduction

Visual stimulus induced gamma oscillations are a prominent feature of the local field potential in the visual cortex, and have been observed in humans [1–3], non-human primates [4–6], cats [7,8] and rodents [9,10], suggesting that they represent a fundamental aspect of the operation of the visual cortex that has been preserved across species. Gamma oscillations are sensitive to the presence of luminance contrast in the visual field [2,5,11–15] and are tuned to properties of luminance-defined gratings, such as their size [16–18], orientation [13,19] and spatial and

**Funding:** The research was funded by a Medical Research Council partnership grant (MR/K005464/ 1) awarded to Krish D Singh. The funders had no role in study design, data collection and analysis, decision to publish, or preparation of the manuscript.

**Competing interests:** The authors have declared that no competing interests exist.

temporal frequency [1,20–25]. However there is some debate about the extent to which they are generated by more naturalistic stimuli [26,27].

While an early study using electrocorticography in macaques reported gamma oscillations to small coloured stimulus patches [6], subsequent studies in humans using magnetoencephalography (MEG) found no significant gamma response to red-green equiluminant gratings [20,23]. Thus, until recently, there has been a lack of consistent evidence for the existence of gamma oscillations in response to chromatic stimulation. However, two recent studies of local field potentials in macaque primary visual cortex [28,29] and a recent electrocorticography (ECoG) study in human visual cortex [30] have demonstrated that pronounced gamma oscillations can be measured in response to colour stimuli.

In the first of the macaque studies, Shirhatti & Ray [28] demonstrated a gamma oscillatory response to large visual displays of uniform colour. The oscillations were shown to be sensitive to both the hue and saturation of the colour and were of particularly high power for long wavelength stimulation (i.e. saturated red hues). In the second macaque study, Peter and colleagues [29] found pronounced gamma oscillations when circular patches of uniform colour were present within the receptive fields of neurons. The oscillations emerged with increasing patch size (being strongest for the largest patch size tested, 6˚), and were sensitive to the colour contrast between the stimulus patches and the surround. In the human ECoG study, Bartoli and colleagues [30] found both broad and narrow-band gamma responses to various stimuli, including a selection of full-screen colour stimuli. Again, as in the macaque studies, they found a strong gamma response to red stimuli.

These findings suggest that, in addition to luminance-defined stimuli, pronounced gamma oscillations may also be produced by colour stimulation. These data also suggest that gamma oscillations are generated in response to large regions of uniform colour, rather than to stimuli that maximise the occurrence of local colour contrast. This might explain the absence of a gamma response in previous MEG studies, which used chromatic gratings rather than spatially uniform stimuli. However, it is also possible that the gamma response to colour simply cannot be measured non-invasively using MEG. This might be the case if the oscillatory response does not show strong coherence across the visual cortex, and thus does not combine to produce a signal that can be measured outside of the head.

We tested this in a sample of twenty participants, by using MEG to measure gamma oscillations in response to full-screen stimulation with four colour hues (red, green, blue and purple) and to luminance-defined gratings for comparison. We were able to demonstrate the presence of gamma responses to the colour stimuli, with the strongest response occurring to the red stimulus, consistent with previous studies.

In a second experiment, conducted in a sample of twelve participants, we tested whether the strong response to red stimulation could be explained by the gamma response being sensitive to the chromatic separation of each stimulus from the preceding display. We presented stimuli that alternated between each of the three pairings of the three primary colours (red, green, blue) at two levels of chromatic separation (defined in the CIELUV colour space). We observed that the gamma response was significantly greater for high, relative to low, chromatic separation, but that the response was not equal for all pairings. Instead, the response was greater for both red-blue and red-green than for blue-green stimulation at each level of separation.

## Materials and methods

### Participants

After giving written consent, twenty volunteers (6 males; mean age: 27.7 years [st. dev: 5.3 years]) participated in Experiment 1 and twelve volunteers (5 males; mean age: 25.5 years [st.

dev: 5.2 years]) participated in Experiment 2. All participants had normal or corrected-to-normal vision (including no self-reported colour blindness).

## Experimental procedure

In both experiments, whole head MEG recordings were made using a CTF 275-channel radial gradiometer system sampled at 1200 Hz (0–300 Hz bandwidth after application of hardware anti-aliasing). One sensor in Experiment 1 and three sensors in Experiment 2 were turned off prior to acquisition due to excessive sensor noise. An additional 29 reference channels were recorded for noise cancelation purposes and the primary sensors were analysed as synthetic third order gradiometers [31].

During data acquisition, participants passively viewed a series of visually presented stimuli (descriptions of stimuli used in each experiment are given below). Stimulus presentation was implemented in MATLAB (The Mathworks, Inc.: Natick, MA, USA) using Psychtoolbox-3 (Brainard, 1997; Kleiner et al., 2007; Pelli, 1997). All stimuli were back-projected onto a screen inside the magnetically shielded room using a Propixx projector (Vpixx Technologies, Inc: Saint-Bruno, QC) running at a 480 Hz refresh rate and a 960 x 540 resolution. The screen measured 51.5 x 29.5 cm and was positioned at a viewing distance of ~55cm. Stimuli were presented at full screen size and therefore subtended a visual angle of approximately 50˚ x 30˚. In each of the two experiments, stimulus presentation was split across two blocks each of approximately 16 minutes length, with participants given a short break between blocks.

Participants were instructed to keep their head as still as possible throughout data acquisition, and head position was monitored continuously using three electromagnetic head localisation coils (placed on the nasion and left and right pre-auricular points).

For source localisation purposes, all participants had a structural MRI scan, which (with one exception, explained below) was acquired on a 3T MAGNETOM Prisma scanner (Siemens Healthcare: Erlangen, Germany) equipped with a 32-channel receive-only head coil. T1-weighted anatomical images were acquired with a three-dimension (3D) magnetisation-prepared rapid gradient-echo (MP-RAGE) at 1 mm resolution and a size of 256 x 256 x 256 voxels. For one participant, the structural scan was instead acquired on a 7T MAGNETOM scanner (Siemens Healthcare: Erlangen, Germany) at 0.65 mm resolution and a size of 320 x 320 x 256 voxels.

All experimental procedures were approved by the ethics committee of the School of Psychology, Cardiff University.

## Stimuli—Experiment 1

In choosing the stimulus conditions for experiment 1, our aim was to replicate the key findings of Shirhatti and Ray [28] with respect to the tuning of gamma power to stimulus hue (at the time of running the experiment we were not yet aware of the research of Peter and colleagues [29] or Bartoli and colleagues [30], and therefore their findings did not influence the choice of stimuli design). We did not have sufficient time to test the full range of 36 hues used by Shirhatti and Ray, so we instead chose four colour hues (defined using the HSV colour model) that we considered to be of most interest based on their data: 0˚ (red) which corresponded to the hue that generated the highest gamma power; 210˚ (blue) which corresponded to the second highest local maxima of gamma power; 270˚ (purple) which corresponded to the hue that produced the weakest gamma power; and 120˚ (green) which corresponded to a broad region of the hue circle that produced intermediate gamma power.

Like Shirhatti & Ray we presented all colours at full saturation, but unlike Shirhatti & Ray we matched physical luminance as closely as possible across conditions (and to baseline

stimulation). The chromaticity (in the CIE xyY colour space) and luminance of each stimulus was measured using an i1 Display Pro colorimeter (X-Rite, Inc: Grand Rapids, MI), and is given in Table 1 (along with the corresponding measurements from Shirhatti and Ray's study).

Following the example of Shirhatti & Ray we also included a luminance-defined grating as an additional condition in order to compare the gamma response to chromatic stimuli with the better-established gamma response to gratings. This condition involved presentation of a static, horizontal, sine grating with mean luminance of 33.3 cd/m$^2$ (matching the luminance of the baseline and colour stimuli as closely as possible), Michelson contrast of 98% and spatial frequency of 2.25 cycles per degree.

Stimuli were each presented for 70 trials (35 trials per block) during which a grey baseline stimulus was presented for a random interval between 1.5 s and 2 s, followed by one of the experimental stimuli for a duration of 1 s (Fig 1). Participants were instructed to continuously fixate on a centrally-presented white circle measuring 0.37˚ that was present throughout the experiment. The order of trials was randomised within each block.

### Stimuli—Experiment 2

Stimulus design in experiment 2 was influenced by the stimuli used by Haigh and colleagues [32] to measure the alpha response to colour stimuli of different chromatic separation. As in their study, we presented stimuli that alternated between pairs of the three primary colours (red, green and blue) with varying degrees of chromatic separation (defined by distance between colours in the CIELUV colour space). However, unlike the Haigh et al. study we tested only two (rather than three) levels of chromatic separation for each pair and used stimuli that were spatially uniform (rather than chromatic gratings).

In order to create our stimuli, we started with the three colour primaries of the projector and adjusted the colours so that they were of (approximately) equal physical luminance and were approximately equidistant from each other and from white in CIELUV colour space. These three modified primaries were then used for the high separation condition (Table 2). In order to create the colours used for the low separation, we found the mid-point in the colour space between each pair of the high separation colours, and then found the colours that were approximately mid-way between each of the modified primaries and the corresponding mid-points. This resulted in three pairs of colours that each differed along the same direction in the colour space as the high separation pairs but separated by only half the distance (Fig 2; Table 2). Note that this meant that the colour hue used in the low separation conditions did not always matched the 'named' hue (e.g. in the low separation red-green pair, the 'red' stimulus had an orange hue, while in the red-blue pair the 'red' was closer to magenta). The chromaticity and luminance of each stimulus was measured using a SpectroCAL Spectroradiometer (Cambridge Research Systems Ltd: Rochester, UK).

**Table 1. Chromaticity and luminance measurements for stimuli used in Experiment 1.**

|  |  | Experiment 1 |  | Shirhatti & Ray (2018) |  |
|---|---|---|---|---|---|
| **Condition** | **Hue** | **Chromaticity (x,y)** | **Luminance (cd/m$^2$)** | **Chromaticity (x,y)** | **Luminance (cd/m$^2$)** |
| Red | 0˚ | (0.68, 0.32) | 33.1 | (0.65, 0.33) | 26.0 |
| Green | 90˚ | (0.15, 0.74) | 33.0 | (0.40, 0.55) | 100.0 |
| Blue | 210˚ | (0.15, 0.21) | 33.2 | (0.23, 0.27) | 50.6 |
| Purple | 270˚ | (0.30, 0.11) | 32.9 | (0.29, 0.13) | 20.0 |
| Baseline (grey) | N/A | (0.30, 0.33) | 33.1 | (0.35, 0.36) | 60.1 |

*The table shows measurements for the four colour stimuli and the baseline stimulus, as well as for the equivalent conditions used by Shirhatti & Ray [28].*

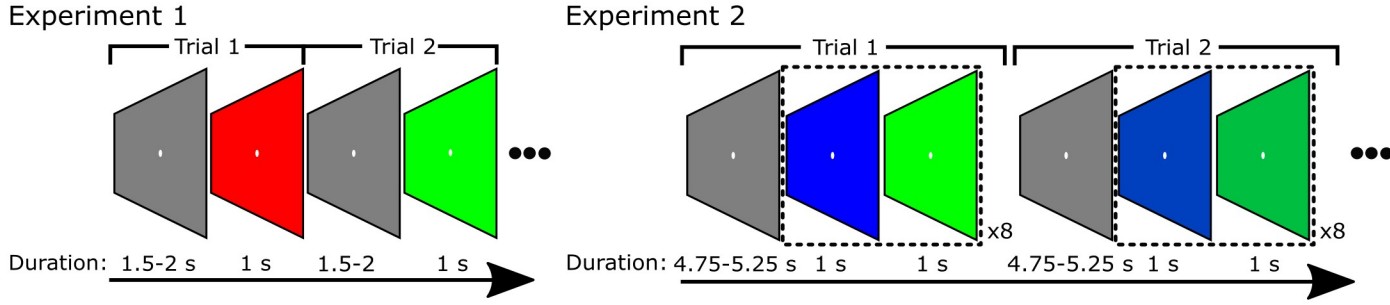

**Fig 1. Illustration of the trial structures used in Experiment 1 and 2.**

**Table 2. Chromaticity and luminance measurements for stimuli used in Experiment 2.**

| | Separation | | | |
|---|---|---|---|---|
| | **Low** | | **High** | |
| **Colour pair** | **Chromaticity (u',v')** | **Luminance (cd/m$^2$)** | **Chromaticity (u',v')** | **Luminance (cd/m$^2$)** |
| Red | (0.32, 0.48) | 33.0 | (0.37, 0.54) | 32.9 |
| Blue | (0.22, 0.34) | 32.8 | (0.17, 0.28) | 32.9 |
| Red | (0.29, 0.55) | 33.4 | (0.37, 0.54) | 32.9 |
| Green | (0.14, 0.56) | 32.6 | (0.05, 0.57) | 33.1 |
| Blue | (0.15, 0.35) | 33.1 | (0.17, 0.28) | 32.9 |
| Green | (0.09, 0.50) | 32.9 | (0.05, 0.57) | 33.1 |

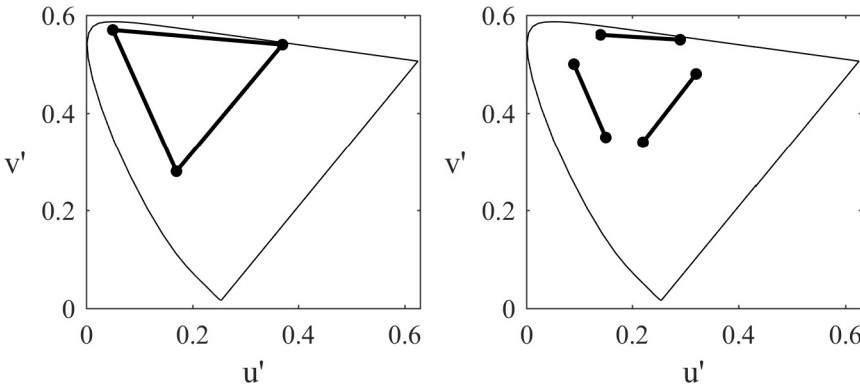

**Fig 2. Chromaticity of the conditions used in Experiment 2.** Left and right panels show the high and low separation conditions respectively plotted on the CIE 1976 uniform chromaticity scale diagram.

Subjects therefore underwent six conditions of stimulation corresponding to the two levels of separation by three colour pairs (red-blue, red-green, blue-green). Stimuli in each condition were presented in a series of 16 trials (8 trials per block) during which the display alternated eight times between the two colours in the pair for 16 s (thus, each colour was displayed in each alternation for 1 s; see Fig 1). The order of colours within the stimulus was counter-balanced across trials within each condition. Each stimulus was preceded by a grey baseline stimulus (luminance: 33.3 cd/m$^2$) presented for a random interval between 4.75 s and 5.25 s. Participants were instructed to continuously fixate a centrally-presented white circle

measuring 0.37˚ that was present throughout the experiment. The order of trials was rando-mised within each block.

## Data analysis

MEG data were acquired as continuous recordings and subsequently segmented into epochs corresponding to the experimental trials. In experiment 1 epochs were defined as -1.25 to 1.25 s relative to stimulus onset. In experiment 2 epochs were defined as -3.5 to 17 s relative to stim-ulus onset. Artefact rejection was performed offline by manually inspecting the data and dis-carding epochs with large muscle or head-movement-related artefacts (because this process is inherently subjective, a single researcher performed artefact rejection in each experiment to avoid inter-rater differences in visual artefact identification). The mean number of rejected tri-als for each participant was 21 in Experiment 1 (maximum number of trials for any individual subject: 121) and 6 in Experiment 2 (maximum: 19).

The locations of the head localisation coils were marked manually on each subject's struc-tural MRI scan, and the MEG and MRI data were co-registered separately for each block based on the mean position of the coils during the block. A multiple local spheres forward model [33] was then derived by fitting spheres to the individual's brain surface extracted from their MRI using FSL's Brain Extraction Tool [34].

Source analysis was then performed using the Synthetic Aperture Magnetometry (SAM) beamformer [35]. Data were bandpass filtered at 25–100 Hz using a fourth-order bi-direc-tional Butterworth filter (this choice of bandwidth was based on data collected from three pilot subjects) and data covariance matrices were subsequently calculated from the concatenation of all trials across all conditions (excluding trials containing artefacts), so that a common set of weights were used across conditions.

For each of the two blocks of acquisition per participant, beamformer images of the paired *t*-statistical difference in source power between baseline (Experiment 1: -1 to 0 s; Experiment 2: -3 to 0 s) and stimulus (Experiment 1: 0 to 1 s; Experiment 2: 0 to 16 s) were created at 4 mm isotropic resolution. The location of the local maximum within occipital cortex was found for each image (where multiple maxima were present with occipital cortex, the maximum closest to the occipital pole was selected). The beamformer weights for this location were then used to generate virtual sensor time series for each trial.

For a single participant in each of the two experiments, the beamformer images did not contain an increase in power in occipital cortex due to the presence of a strong beta power decrease throughout the brain volume that masked the gamma response. In both cases we generated new images at 40–100 Hz bandpass in order to identify the location of the largest positive t-statistical difference in the medial occipital cortex.

Time-frequency spectrograms of each participant's virtual sensor time series were pro-duced for each condition by bandpass filtering with a series of filters centred in 1 Hz steps from 15 to 110 Hz using fourth-order bi-directional Butterworth filters with a bandwidth of 8 Hz. The Hilbert transform was then used to compute the envelope of the analytic signal at each frequency and the instantaneous amplitude of the envelope at each time sample was aver-aged across trials. The resulting spectrograms were baseline corrected by calculating the per-centage change in amplitude at each time-frequency sample relative to average amplitude for the corresponding frequency over the baseline time period (Experiment 1: -1 to 0 s; Experi-ment 2: -3 to 0 s).

In Experiment 2 we also created separate spectrograms of the response to each colour within a pair by averaging the spectrogram from 0 to 1 s relative to each presentation of each colour within each pair, both within and between trials. We excluded the first 1 s from each

trial from this averaging process, in order to exclude the initial stimulus onset response from these spectrograms.

All statistical comparisons between conditions were performed using standard parametric test statistics, but as we could not assume normality of the data, all *p*-values were determined using permutation testing: each test statistic was compared with the distribution of 99,999 randomly selected within-subject permutations. Tests were considered significant if $p < 0.05$ (the minimum *p*-value given the number of permutations was 1e-5).

Calculation of the forward model, covariance matrices and beamformer weights were performed using propriety software supplied by the MEG manufacturer (CTF). Subsequent analysis was performed in MATLAB using a combination of the Fieldtrip toolbox [36] and custom scripts.

## Results

### Experiment 1

The purpose of our first experiment was to establish whether we could record the gamma response to colour stimulation using MEG. As it is already well established that the gamma response to luminance contrast can be measured with MEG [2,14,15], we also sought to compare the gamma response to colour stimuli with that of luminance-defined gratings.

For each subject, we generated spectrograms of the virtual sensor time-series of the response to four colour hues (red, green, blue and purple) and to gratings (group averages shown in Fig 3). The group average spectrogram of the response to gratings shows the 'classic' gamma response: a narrow-band response within the gamma frequency that is sustained for as long as the stimulus is present (although gradually decreasing in both amplitude and frequency over time). A similar component is evident in the response to colour stimuli, albeit of much weaker amplitude. Two other responses can also be observed in the spectrograms for colour stimuli: a lower frequency response around 30 Hz that emerges gradually over time, and a higher frequency response around 85 Hz that appears to be present only for the red stimulus and which is of a particularly high amplitude immediately after stimulus onset. We will refer here to these three distinct spectral components as the low frequency, mid frequency and high frequency gamma responses.

In order to better visualise the spectral properties of the sustained part of the gamma responses we averaged the spectrograms across time from 300–1000 ms (Fig 4). Although the sustained part of the response is clearly not stationary, the change is sufficiently gradual for these group average spectra to be broadly representative of the sustained response. As observed in the spectrograms, the mid frequency gamma response is stronger to gratings than to the colour stimuli. This differs from the findings of Shirhatti & Ray [28] who observed in their data that the gamma response to some colour hues (particularly those around the red part of the spectrum) could have a greater amplitude than the response to gratings.

Despite the relatively weaker responses to colour, the spectra of the response to each colour have two clear spectral peaks corresponding to the mid and low frequency gamma responses (as well as a smaller peak corresponding to the high frequency response to the red stimulus). The frequency of both responses tended to be higher for the colour than the grating stimuli. The exception to this was the mid frequency response to the green stimulus which was lower in frequency than the equivalent response to the other colour stimuli (in Fig 4 the mid frequency response to the green stimulus corresponds to the 'bump' around 47 Hz in the spectrum).

As we could not accurately measure the peak amplitude in all cases, we instead measured the amplitude of the two oscillations by averaging the spectra within the frequency ranges of 25–40 Hz and 40–70 Hz for each subject in each of the four colour conditions. We calculated

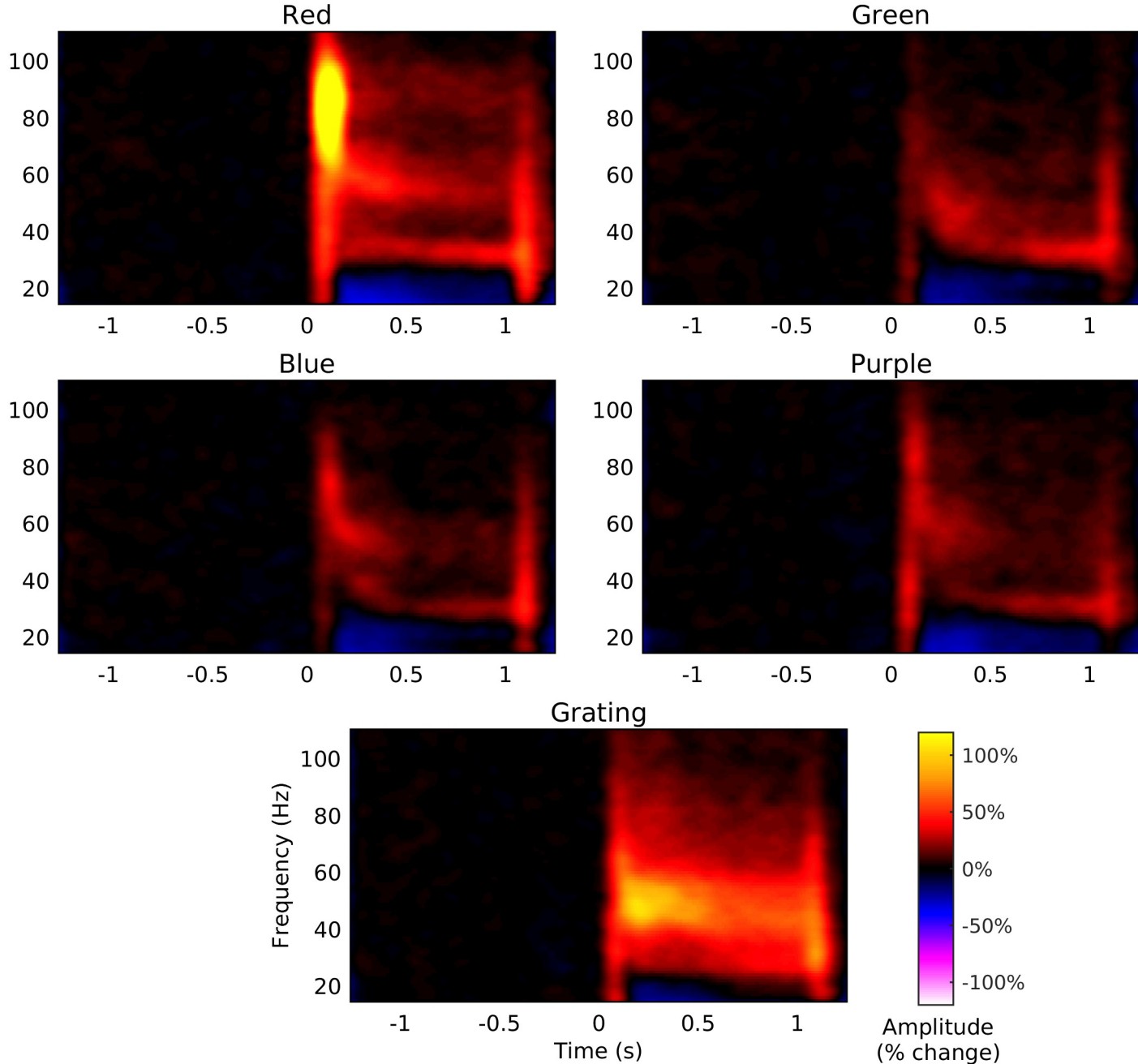

**Fig 3. Group average spectrograms of virtual sensor responses to each stimulus.** The colour scale for each spectrogram shows amplitude as percentage change from baseline.

the same measures for the grating condition but used the frequency ranges of 20–35 Hz and 35–65 Hz due to the lower frequency of the responses to the grating stimulus. The mean amplitude of the response in each of the three condition for the low and mid-frequency range are given in Table 3.

Tests of within-subject differences revealed that both mid frequency ($F_{(4,76)} = 60.6$, $p = 1e-5$) and low frequency ($F_{(4,76)} = 7.5$, $p = 5e-5$) amplitude were significantly different

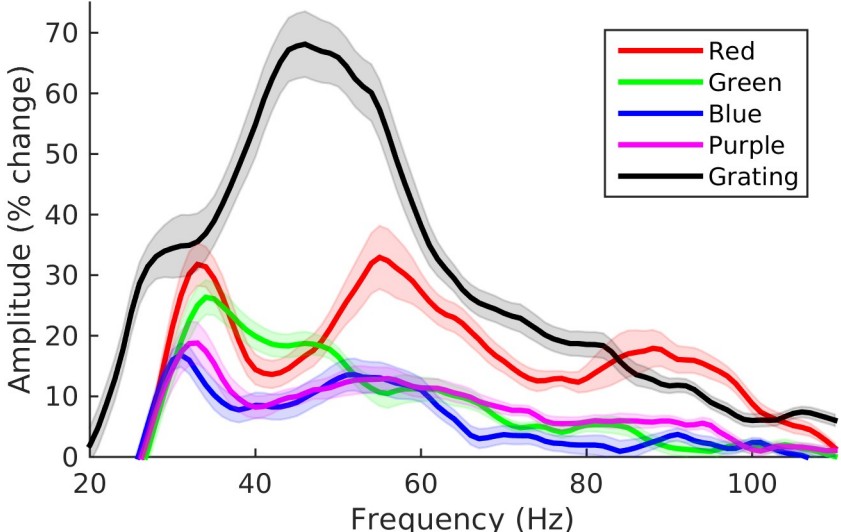

**Fig 4. Group average spectra of the sustained (300–1000 ms) response to each stimulus.** Plot for each condition shows mean amplitude as percentage change from baseline (+/- within-subject standard error).

between conditions. To determine which conditions differed, we performed a series of paired *t*-tests between each pairwise combination of conditions, controlling the familywise error rate at 0.05 using the Holm-Ŝidāk method (Table 4). For mid frequency amplitude all pairs of conditions differed significantly apart from blue and purple. In the low frequency range, the grating condition had significantly greater amplitude than the colour conditions, and the red condition significantly greater amplitude than the blue and purple conditions, but no other pairs of conditions differed. Thus, in agreement with previous studies, we found a stronger response to red stimulation than to other colours.

The timecourse of the low and mid frequency gamma responses was broadly similar across conditions (Fig 5), with high amplitude transients occurring shortly after stimulus onset and offset and a lower amplitude sustained response that occurs while the stimulus is present. The sustained response tends to increase gradually over time in the low frequency range but conversely tends to decrease over time in the mid frequency range. Notably, the onset transient to the green stimulus in the mid frequency range was delayed relative to the transients generated by the other stimuli, adding further evidence to the suggestion that there are differences in the time-frequency properties of the mid frequency gamma response to green stimulation relative to the other stimuli tested.

Fig 3 suggests that the high-frequency gamma response was most prominent around stimulus onset, rather than in the sustained part of the response. Therefore, to better visualise this

**Table 3. Group mean amplitude (measured as percentage change from baseline) and within-subjects standard error for each condition for each frequency band in Experiment 1.**

| Condition | | Red | Green | Blue | Purple | Grating |
|---|---|---|---|---|---|---|
| Low frequency | Mean (%) | 18.2 | 15.8 | 9.5 | 10.6 | 25.8 |
| | Std Err (%) | 2.4 | 1.7 | 1.7 | 2.3 | 3.5 |
| Mid frequency | Mean (%) | 22.3 | 13.3 | 9.3 | 10.8 | 52.4 |
| | Std Err (%) | 2.1 | 1.3 | 1.9 | 1.5 | 3.9 |

**Table 4. Test statistics and p-values for all pairwise comparisons between conditions in Experiment 1.**

| | | | | Green | Blue | Purple | Grating | Low frequency | |
|---|---|---|---|---|---|---|---|---|---|
| Red | t (19) | 3.8 | | 0.8 | 5.8 | 4.9 | 7.5 | t (19) | Red |
| | p | 0.0002* | | 0.5 | 0.006* | 0.004* | 0.0001* | p | |
| Green | t (19) | 16.7 | 3.3 | | 2.6 | 3.8 | 9.6 | t (19) | Green |
| | p | 0.00001* | 0.004* | | 0.02 | 0.03 | 0.0001* | p | |
| Blue | t (19) | 16 | 6.5 | 1.6 | | 0.5 | 11.2 | t (19) | Blue |
| | p | 0.00001* | 0.00001* | 0.12 | | 0.6 | 0.0002* | p | |
| Purple | t (19) | 60.6 | 76.4 | 78.7 | 9.2 | | 3.3 | t (19) | Purple |
| | p | 0.00001* | 0.00001* | 0.00001* | 0.00001* | | 0.005* | p | |
| Mid frequency | | Green | Blue | Purple | Grating | | | | |

The top right part of the table shows results for the low frequency gamma response and the bottom left part of table shows the results for the mid frequency gamma response. Starred values indicate effects that are significant after correction for multiple comparisons.

response, we averaged the spectrograms within the time range 50–200 ms in order to generate spectra of the stimulus onset response (Fig 6). A clear high amplitude peak is present in the group average spectrum of the response to the red stimulus at approximately 85 Hz but does not appear to be present for the other stimulus conditions. One subject exhibited a high-frequency response to red stimulation of such high amplitude that the onset part of this response was visible in their raw data (Fig 7A). The data exhibits an oscillation appearing briefly after stimulus onset, indicating that the spectral peak represents a true oscillatory response and is not merely a spectral component of the stimulus-onset evoked response. In this subject we performed a beamformer analysis of this onset response (contrasting 65–100 Hz band-limited power from 75–150 ms with power in the baseline period from -400 –-325 ms) which demonstrated that the response originates in medial occipital cortex (Fig 7B).

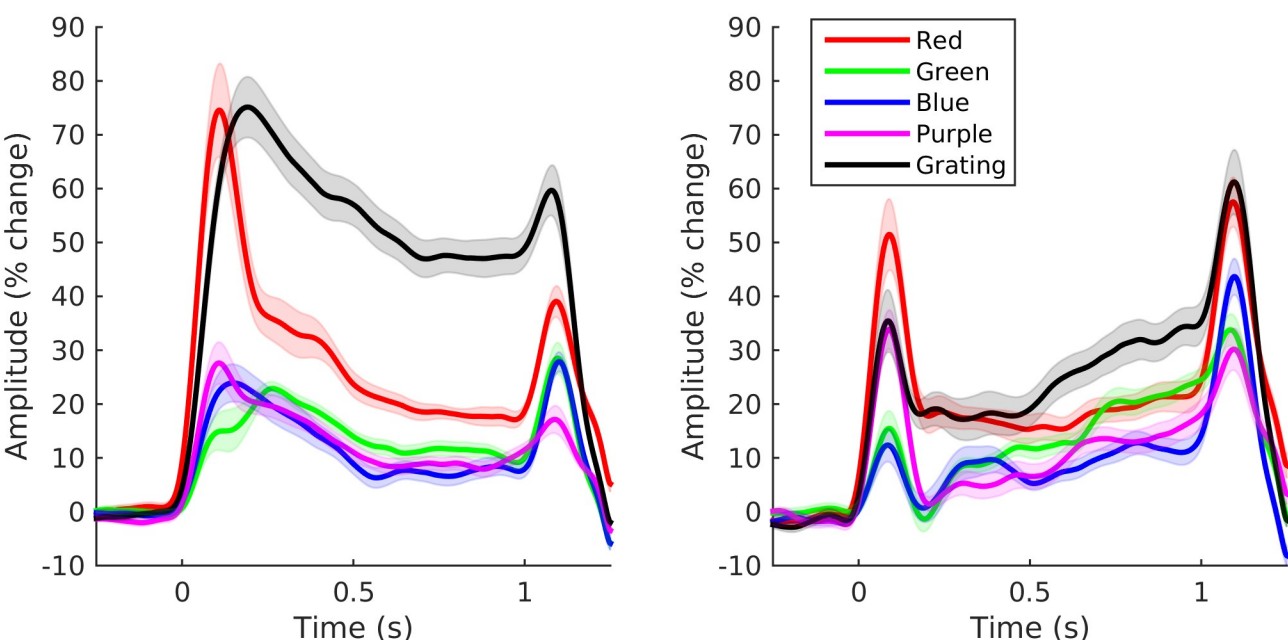

**Fig 5. Group average time series of the sustained low and mid frequency gamma responses to each stimulus.** Plot for each condition shows mean amplitude as percentage change from baseline (+/- within-subject standard error).

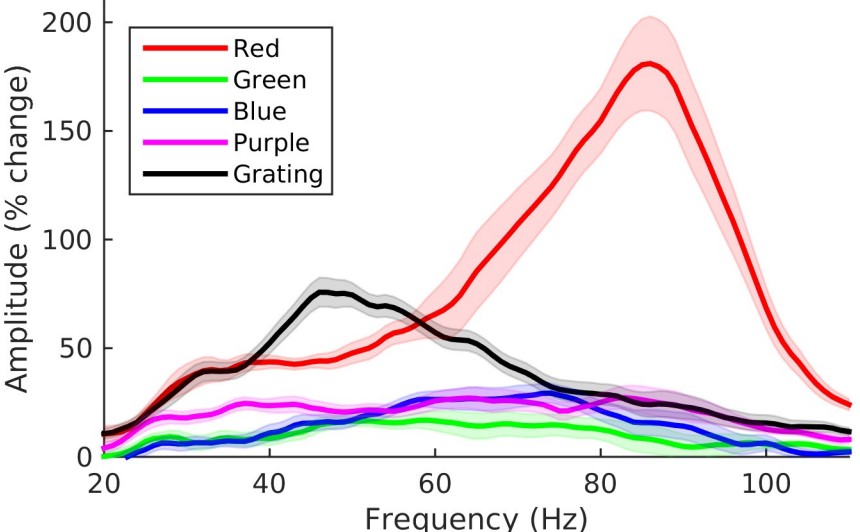

**Fig 6. Group average spectra of the onset (50–200 ms) response to each stimulus.** Plot for each condition shows mean amplitude as percentage change from baseline (+/- within-subject standard error).

## Experiment 2

In experiment 1 we demonstrated that the gamma response to colour could be measured with MEG, and that (consistent with previous studies) the strongest response was induced by red stimulation. We were interested in understanding why red stimulation produced a particularly strong gamma response. We note that Haigh and colleagues [32] recently measured the alpha response to reversing chromatic gratings and found an increased response (in the form of a greater stimulus-induced decrease of alpha power) with increasing chromatic separation of the

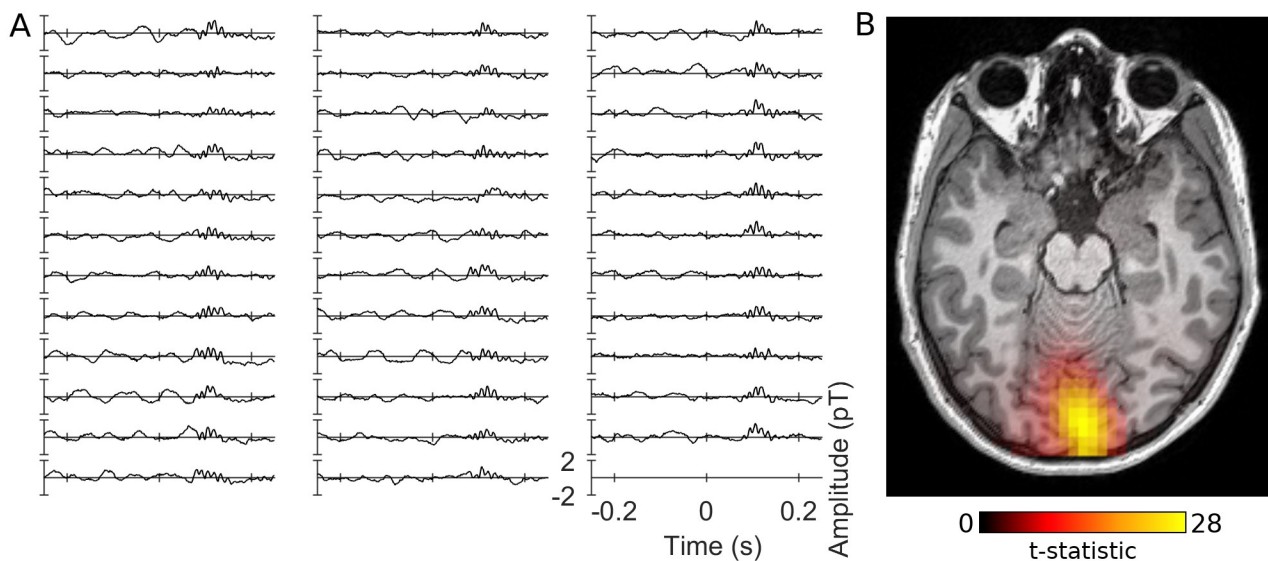

**Fig 7. Data from the subject with the strongest high frequency gamma response to red stimulation.** A) Half a second of unfiltered data (centred on stimulus onset) from a single occipital sensor (MRO42) from each of the 35 trials of red visual stimulation from the first block of the experiment. B) Beamformer localisation of this response.

gratings. When we represented the chromaticity of our gratings in the CIELUV colour space used by Haigh and colleagues, we found that our red stimulus had the greatest chromatic separation from the baseline stimulus (Fig 8). Thus, our enhanced response to red may not be intrinsic to red hues, but simply due to the chromatic separation of that stimulus from baseline. To test this possibility, we ran a second experiment following a similar design to that used by Haigh and colleagues. We presented subjects with full-screen colour stimuli that altered between pairs of the three primary colours (red, green, blue) at two level of chromatic separation (we refer to these two levels here as low and high separation). The coordinates of these stimuli in the CIELUV colour space are shown in Fig 2.

Fig 9 shows the spectrograms of the gamma response for each of the six conditions of colour pairing and separation. In each spectrogram we can see a clear gamma response following the onset of each colour change (occurring every 1 s from 0 s), although the responses in the low separation conditions are much weaker than the responses for high colour separation.

To better quantify this effect, we separated the gamma response into three frequency bands —low (25–40 Hz), mid (40–70 Hz) and high (70–100 Hz)–based on the data from Experiment 1. We then calculated the average amplitude for each condition within a time window of 0–16 s (Fig 10) and analysed this data as a two-factor repeated-measures design for each frequency band (Table 5).

We found that the difference between the high and low separation conditions was significant for all three frequency bands. We also found a significant main effect of colour pair at all three bands, which reflected the fact that the two colour alternations containing red generated a stronger gamma response than the blue-green stimulus. In the mid and high frequency

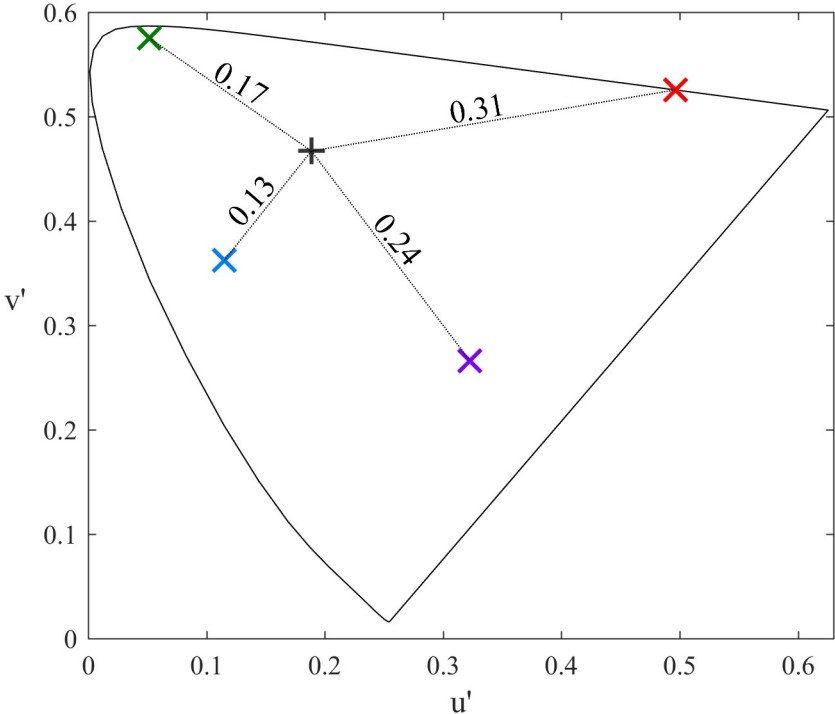

**Fig 8. Chromaticities of the colour stimuli used in Experiment 1.** Coloured crosses show the chromaticities of the four colour stimuli plotted on the CIE 1976 uniform chromaticity scale diagram. The black + symbol shows the chromaticity of the baseline stimulus. Dashed lines and associated numbers show the chromatic distance of each stimulus from baseline.

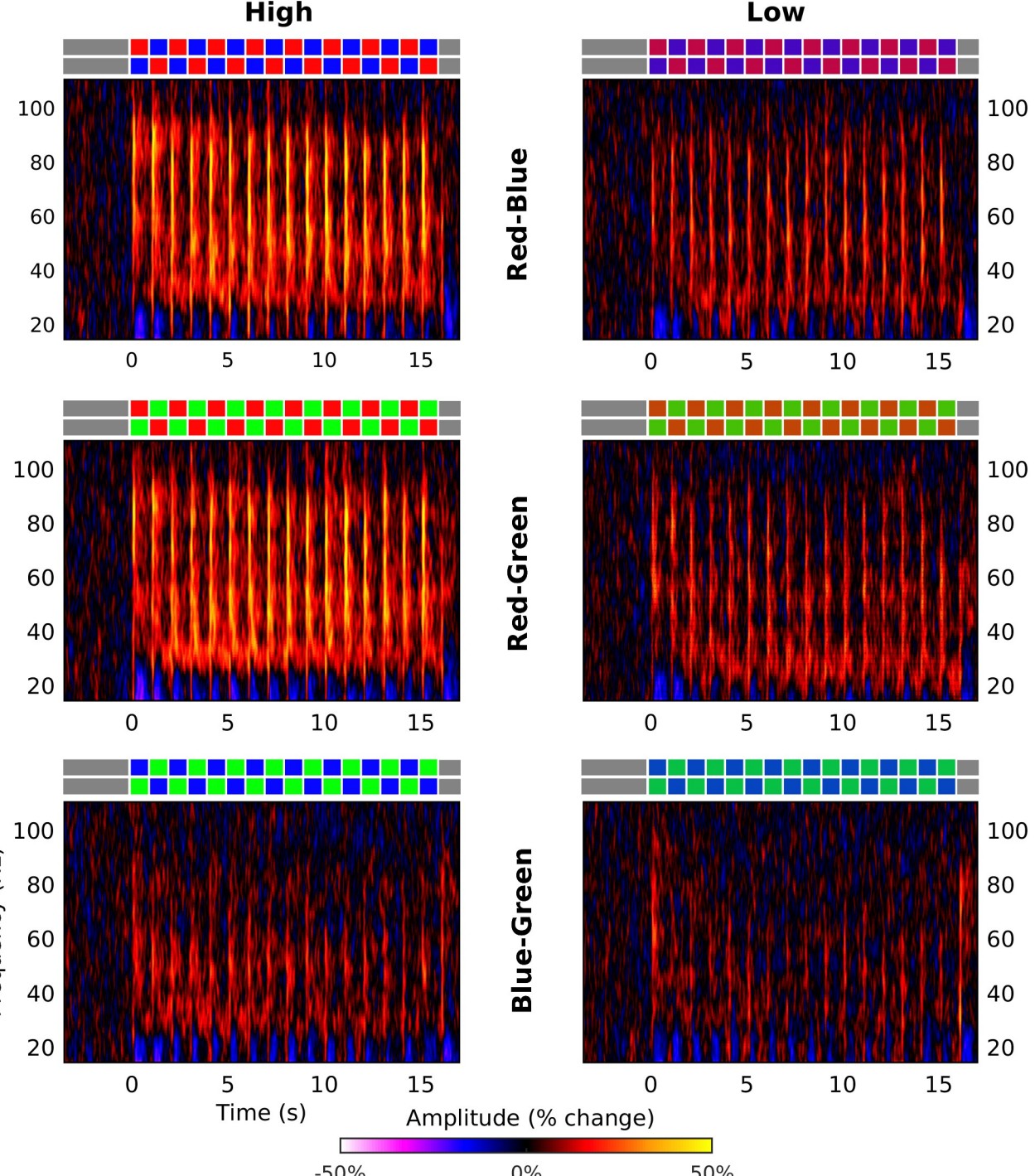

**Fig 9. Group average spectrograms of virtual sensor responses to each stimulus.** Rows correspond to each colour pair while columns correspond to the two levels of colour separation. The colour scale for each spectrogram shows amplitude as percentage change from baseline. The coloured patches above each spectrograms illustrate the two sequences (corresponding to the two counter-balanced stimulus orders) of display colours that were presented in each condition.

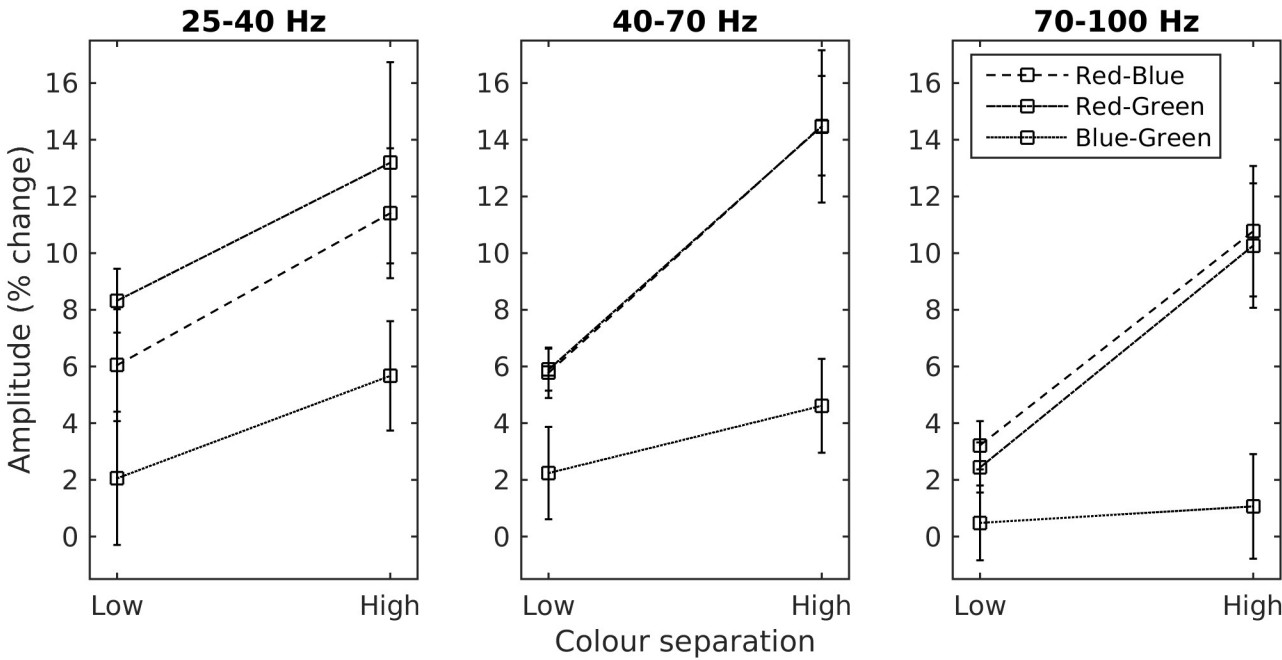

**Fig 10. Group amplitude of the gamma response.** Each panel shows the mean amplitude (+/- within-subject standard error) of each of three frequency bands of the gamma response for each condition.

bands there was also a significant interaction between colour pairing and separation, reflecting the fact that not only was there a weaker response to the green-blue stimulus, but that that response showed a smaller increase with increasing colour separation.

The order of colours was counter-balanced across trials, meaning that the response for each cycle of alternation shown in Fig 10 is the average response across both colours within each pair (for instance, in the red-blue conditions half of the trials followed a cycle of red-blue-red-blue. . . while the other half of the trials followed a cycle of blue-red-blue-red. . . and therefore the response to the red and blue stimuli are averaged together in the average spectrograms). In order to better understand the differences in the response to each colour within each pair, we generated separate spectrograms of the response to each colour in each pair for the high separation condition (Fig 11). These spectrograms demonstrate that the colour pairings containing red produced a stronger gamma response for two reasons. Firstly, as observed in experiment 1, the gamma response to red stimulation was of higher amplitude than to stimulation with blue or green. Secondly, the gamma response to the blue and green stimuli was greater when those

**Table 5. Test statistics and p-values for the main effects and interactions in Experiment 2.**

| | | Frequency Band | | |
|---|---|---|---|---|
| **Contrast** | | **25–40 Hz** | **40–70 Hz** | **70–100 Hz** |
| Colour separation | F (1,11) | 6.9 | 15.4 | 14.4 |
| | *p* | *0.002* | *0.00002* | *0.00002* |
| Colour pair | F (2,22) | 8.3 | 35.5 | 24.6 |
| | *p* | *0.01* | *0.00002* | *0.0001* |
| Separation x pair | F (2,22) | 0.3 | 9.6 | 8.3 |
| | *p* | *0.73* | *0.0005* | *0.0004* |

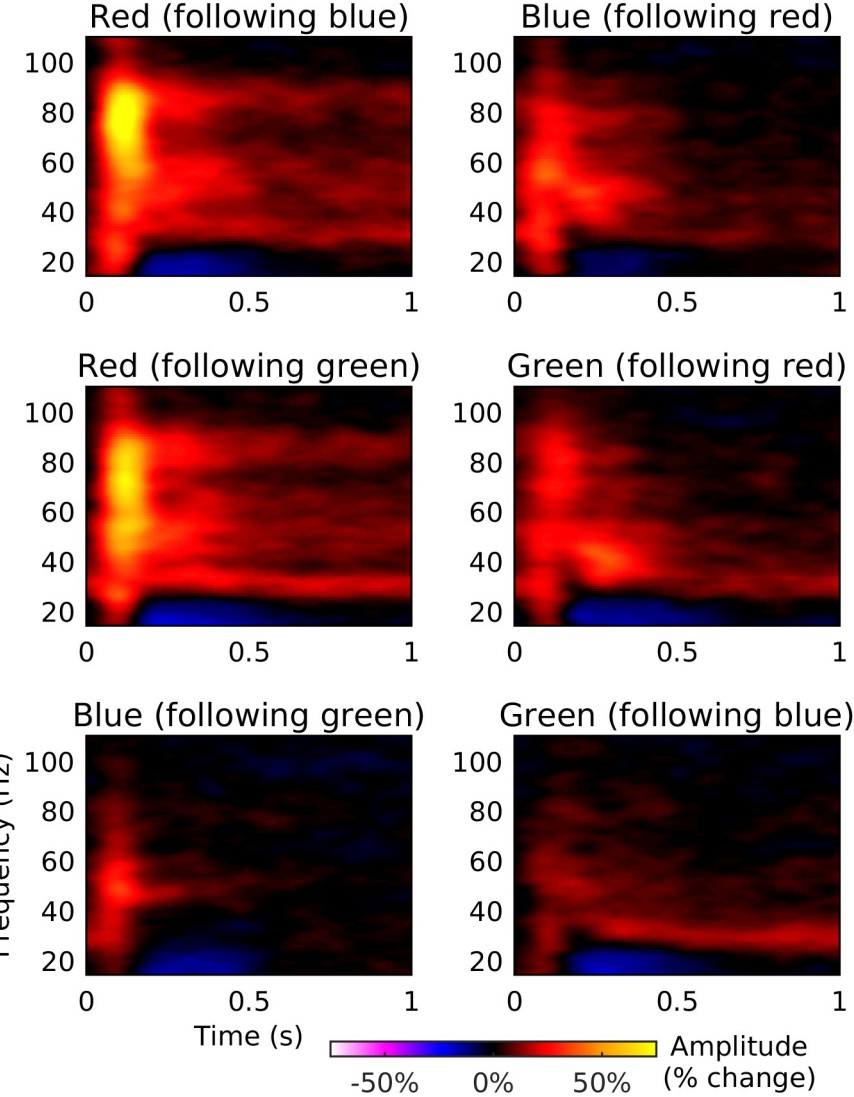

**Fig 11. Group average spectrograms of response to each colour within each pair.** Rows correspond to each stimulus pair. The colour scale for each spectrogram shows amplitude as percentage change from baseline.

stimuli were paired with a red stimulus than when they were paired with each other. Thus, not only did the onset of the red stimulus produce the strongest gamma response, but the offset of the red stimulus appeared to enhance the response to the subsequent stimulus.

## Discussion

It has been recently demonstrated in both macaques [28,29] and humans [30] that gamma oscillations can be generated in visual cortex in response to large colour stimuli (either full-screen stimulation or large colour 'patches' extending several degrees). Long wavelength stimuli (i.e. of red hue) produce a particularly strong gamma response. In our first experiment we have shown that the same oscillations can be measured using MEG and that, consistent with previous studies, red stimuli produce the strongest gamma response.

In our second experiment we tested whether this enhanced response to red stimulation might be due to the greater chromatic separation of the red stimulus from the baseline

stimulus in Experiment 1 (in line with the finding of Haigh and colleagues [32], that the alpha response to chromatic stimuli increases with increasing chromatic separation). We were able to demonstrate that the gamma response to alternating colour stimuli increased with increasing chromatic separation between the stimuli. However, this effect alone could not explain the enhanced response to red stimuli. Firstly, there was a stronger response to the two pairs containing red than to the blue-green pairs, despite chromatic separation being matched between the pairs. Secondly, if the response was purely determined by chromatic separation, we would expect the response to each colour within a pair to be equal, but it is evident in Fig 11 that the response was greater for the red stimulus within both the red-blue and red-green conditions.

## The enhanced response to red stimulation

In considering why the red stimuli produced a greater gamma response than the other colour hues it is important to note that, following Haigh and colleagues [32], we measured chromatic separation based on coordinates in the CIELUV colour space. However, while this space is designed to achieve perceptual uniformity, it does not map directly on to the physiological representation of colour in early visual cortex. It therefore may be the case that differences in response between hues is due to chromatic separation but measured using a different colour space, for instance based on the physiological mechanisms of colour opponency (e.g. the DKL colour space).

Interestingly, when we measured the colour opponency (based on the cone contrast between each of pair of colours) of the stimuli used in Experiment 2, we found that colour modulation for the red-green pair predominantly involved L-M opponency, while for the blue-green pair it mainly involved S-(L+M) opponency (for the red-blue pair both colour opponent channels were modulated). This might suggest that the gamma response is tuned primarily to colour modulations along an L-M opponent axis, which would be consistent with evidence that colour responsive cells in primary visual cortex generally do not receive strong S cone inputs, and are therefore mainly sensitive to L-M opponency [37]. This could explain why the response to the green and blue hues was stronger when preceded by a red hue, as this would have produced stronger temporal modulation of L-M opponent processes, whereas the green-blue pair would have produced very little L-M opponent response.

However, even if we assume that the colour space we used did not accurately reflect the colour space of the underlying neuronal representation, the fact that red produced a stronger gamma response even within colour pairs suggests that this cannot be purely explained by chromatic separation alone. We note here that Peter and colleagues [29] were able to abolish the advantage for red stimuli by shifting to a black, rather than grey, baseline stimulus. Based on modelling of their data, they attributed this to M cones showing greater adaptation to the grey baseline stimulus than L & S cones. Our data did not allow us to directly test this hypothesis, but like Peter and colleagues we found in Experiment 1 that the frequency of the gamma response was reduced for the green stimulus compared to the other hues, which might point to the dynamics of the cortical response being different for medium wavelength hues.

## Differences to previous studies

Although our data replicated the gamma response to colour hues seen in previous studies, there were also several important differences.

One major difference was the magnitude of the gamma response to the grating stimulus relative to the colour stimuli. One of the most striking findings in Shirhatti & Ray's [28] study was that some colour hues (particularly red hues) produced a gamma response of a much greater magnitude than the response to luminance-defined gratings. By contrast, we measured

a gamma response to gratings that was far in excess of that to any colour hue. This was true not only in the group average, but also for most (but not all) individual subjects. There are several possible explanations for this finding.

Firstly, this may reflect the fact that the colour hues used here did not exactly match those used by Shirhatti & Ray, due to the use of a different display device. However, we note that the red hue used in this study was relatively close to that used by Shirhatti & Ray (see Table 1 for comparison) and thus we might expect it to have produced a similarly strong response in this study.

Alternatively, this result could reflect an inter-species differences between humans and macaques. Consistent with this, our work more closely matches Bartoli and colleagues' [30] ECoG study in humans, where the (narrow-band) gamma response also tended to be stronger for gratings than for colour stimuli.

A third possibility is that this difference in findings might reflect a difference in the spatial scale of measurements made in invasive recordings of local field potentials (LFPs) versus those made by MEG (and ECoG). Oscillations present in LFP recordings reflect coherent potentials sampled from within a few mm$^2$ of the tip of the electrode, whereas oscillations measured using MEG must be coherent over at least several mm$^2$, and perhaps as much as several cm$^2$. Furthermore, it is known that gamma oscillations to luminance-defined gratings show some degree of coherence across several mm of cortex [17], creating the conditions necessary for the signal to be measurable by MEG. It is therefore possible that, if the gamma response to colours exhibited a weaker degree of spatial coherence, a strong signal could be measurable locally within the LFPs but would not combine to produce a similarly strong signal when measured with MEG. This hypothesis could be tested in future by comparing the degree of spatial coherence of LFPs across the cortex for colour versus grating stimuli.

The other major difference between this and the three previous studies is the presence of two additional frequency components of the gamma response, a lower frequency response around 25–35 Hz and a higher frequency response around 70–100 Hz that was present for the red stimulus only. A higher frequency response was present in the studies of Shirhatti & Ray [28] and Bartoli and colleagues [30], but in both cases was described as being a harmonic of the primary gamma response. A similar high frequency response is also evident in Fig 5 of Peter and colleagues [29] but does not appear to have been commented on by the authors. In our data, the high frequency and mid frequency components of the gamma response are not related by any integer multiple of frequency, and therefore the two responses are unlikely to be harmonically related. We suggest instead that these are distinct oscillatory responses.

For some subjects the onset component of this high frequency response had an amplitude several times greater than the average baseline amplitude, and in one subject was powerful enough to be visible in the raw sensor traces. To our knowledge, this is the highest amplitude oscillatory response in the gamma-band yet observed to visual stimulation using MEG. This suggests that the oscillatory currents that generate this response are either locally very high in amplitude or highly coherent across an extended region of the primary visual cortex, and in either case might be expected to have some important functional impact on the cortical response to the inducing stimulus. The underlying mechanisms and functional effects of this response therefore merit further investigation.

By contrast, there is no clear evidence in any of the previous studies for the presence of the low frequency gamma response. We do not have an explanation for why this difference might exist, and it remains to be explored in future studies why a low frequency component appears in this study but not others.

Finally, our findings also differ from two previous MEG studies that have attempted to measure gamma oscillations to chromatic gratings [20,23]. Both studies failed to find a clear

gamma oscillatory response, but not only did they use stimuli substantially smaller than those used here, but they used equiluminant red-green gratings, rather than stimuli of uniform colour hue. This suggests that large regions of homogeneous colour, rather than spatial variation in colour, may be necessary to generate a clear gamma response to chromatic stimuli. This in turn might suggest a link between the gamma response to chromatic stimuli and a class of 'colour-preferring' neurons in primary visual cortex that are known to have low-pass spatial tuning [37–39]. Future studies into the tuning of the gamma response to spatial frequency, particularly at frequencies below 0.5 cycles per degree (the lowest spatial frequency tested in the two previous studies) would help to clarify this issue.

## Implications for photosensitive epilepsy

Finally, one area of research where the current results may have some bearing is in the investigation of the mechanisms of photosensitive epilepsy. On the basis of the tuning of the gamma response to luminance-defined stimuli it has been proposed that gamma oscillations may play a role in the generation of photosensitive seizures [40] and it has been shown that there is a tendency of individuals with a diagnosis of photosensitive to show a heightened gamma response to luminance-defined gratings [41]. However, it is also well known that a photoparoxysmal response can be triggered by coloured flicker, particularly were that flicker involves red [42–45].

Parra and colleagues [46] investigated the frequency at which stimuli alternating between different colour pairs trigger a photoparoxysmal response in individuals with photosensitive epilepsy. They found that while red-blue and red-green flickers could be highly epileptogenic, blue-green flicker triggered far few epileptic discharges. Thus, our data suggest that the stimulus tuning of the gamma response to colour may match that of the stimulus tuning of the photoparoxysmal response, and this therefore leads further weight to a link between visual gamma and photosensitive epilepsy. It should be noted, however, that Parra and colleagues found colour flicker to be most epileptogenic when presented at much higher temporal frequencies (10–15 Hz) than those used in this study. Therefore, it would be useful in future research to measure the tuning of the gamma response to the temporal frequency of colour stimulation.

## Author Contributions

**Conceptualization:** Gavin Perry, Nathan W. Taylor, Philippa C. H. Bothwell, Georgina Powell, Krish D. Singh.

**Formal analysis:** Gavin Perry, Nathan W. Taylor, Philippa C. H. Bothwell.

**Investigation:** Gavin Perry, Nathan W. Taylor, Philippa C. H. Bothwell, Colette C. Milbourn.

**Supervision:** Krish D. Singh.

**Visualization:** Gavin Perry.

**Writing – original draft:** Gavin Perry.

**Writing – review & editing:** Nathan W. Taylor, Philippa C. H. Bothwell, Colette C. Milbourn, Georgina Powell, Krish D. Singh.

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
