## [Decision Letter · Decision Letter 0]

3 Sep 2020

PONE-D-20-14792

The gamma response to colour hue in humans: evidence from MEG

PLOS ONE

Dear Dr. Perry,

thank you for submitting your manuscript to PLOS ONE. After careful consideration and based on the comments of two experts in the field, we feel that it has merit but does not fully meet PLOS ONE’s publication criteria as it currently stands. Therefore, we invite you to submit a revised version of the manuscript that addresses the points raised during the review process.

While revising the manuscript, please respond to the Reviewers's comments in a point-by-point manner and make sure to specifically and thoroughly address the technical and statistical aspects of your work (including Figures) as mentioned by both Reviewers. Please also improve the description of Experiment 2.

Please submit your revised manuscript within six months from this date as otherwise a revision has to be considered a new submission. If you will need more time than this to complete your revisions, please reply to this message or contact the journal office at plosone@plos.org. Please include the following items when submitting your revised manuscript:

We look forward to receiving your revised manuscript.Thank you for choosing PLOS ONE for reporting your research.

Best regards,

Alexander N. 'Sasha' Sokolov, Ph.D.

Academic Editor

PLOS ONE

Journal Requirements:

3.  We note you have included a table to which you do not refer in the text of your manuscript. Please ensure that you refer to Table 3 in your text; if accepted, production will need this reference to link the reader to the Table.

Reviewers' comments:

Reviewer's Responses to Questions

**Comments to the Author**

1. Is the manuscript technically sound, and do the data support the conclusions?

Reviewer #1: Yes

Reviewer #2: Partly

2. Has the statistical analysis been performed appropriately and rigorously? 

Reviewer #1: Yes

Reviewer #2: No

3. Have the authors made all data underlying the findings in their manuscript fully available?

Reviewer #1: Yes

Reviewer #2: Yes

4. Is the manuscript presented in an intelligible fashion and written in standard English?

Reviewer #1: Yes

Reviewer #2: Yes

5. Review Comments to the Author

Reviewer #1: The study of Perry et al is devoted to the MEG gamma responses induced by color hues. The recent invasive studies in animals and intracranial studies in humans have shown that homogeneous color surfaces induce gamma oscillations in visual cortex. This study is the first one that shows that the similar gamma responses can be recorded noninvasively in humans using MEG. The authors also investigated how the gamma response to the hue change is affected by chromatic separation between the colors (in perceptual color space). They confirmed the previously shown ‘preference’ for the red hue. They also described different gamma frequency components induced in MEG by hues.

This is an interesting and useful study. I think that the methods and results are clear. I only have minor comments and suggestions.

1. Please, describe what software has been used for the source localization analysis.

2. Line 199. What is the ‘approximate area of medial occipital cortex’? Please provide more information.

3. It might worth to show the ‘color separations’ used in the 2nd experiment in a figure similar to Fig.6.

4. The ‘red’ color in the low separation condition is not really red, but rather orange. Could it be better to use the word ‘reddish’, ‘bluish’. …?

5. It took me some time to understand the design of the 2nd experiment. To the convenience of the lazy reader, I would suggest to insert a color line that illustrates the sequence of color stimuli above each panel of the Figure 7.

6. In the figure 9 I would suggest to change ‘Red’ to ‘Red-after-Blue’, ‘Red-after-Green’, etc. to improve readability of the figure.

7. Do you have an idea why the offset of the red hue promotes gamma response to the green and blue hues? This worths mentioning in the Discussion section.

8. Figure 5.

A) Traces show one second of unfiltered data (centred on stimulus onset) from a single occipital sensor (MRO42) from each of the 35 trials of red visual stimulation from the first block of the experiment.

It is unclear what ‘from each of the 35 trials’ refers to. It looks like the left panel (A) shows only one epoch…

B) Please provide units for the scale.

9. Line 404. …. we found in Experiment 1 that the frequency of the gamma response was reduced for the green stimulus compared to the other hues,…

The frequency of the gamma peak for the green hue does not seem to be reduced (Fig. 3). The ‘low gamma’ for the green seems have even higher frequency than that for other hues.

Reviewer #2: Perry and colleagues utilized MEG measurements in human subjects to study the influence of stimulus color on visual gamma activity. In their first experiment, the authors show color effects consistent with recent monkey and human studies – i.e. gamma responses occur to uniform color stimuli, which were maximal red stimuli. In the second experiment, the authors employed three primary colors at two different chromatic distances to evaluate the impact of the preceding color presentation on the recorded response. The authors report that the chromatic separation between pairs of stimuli has an effect on the gamma response, although this effect is not equal across different color-pairs, showing a bias for pairs of colors involving red. This latter result is interesting and in line with evidence that the bias for red color can be partially related to adaptation effects although there is more to be addressed for understanding the origin of such a bias. Overall, the manuscript provides an important follow up to recent work and will be of interest to many. Below I highlight a few points of improvement related to the quantification and presentation of findings.

Major comments:

-Overall the quantification of observed effects could be improved. Results for experiment 1 seem to be mostly reported in Figure plots with modest statistical reporting. For example, what were the specific mean responses to each color? Saying post-hoc tests satisfied a corrected p-value is limited, provide the test statistic and related mean values.

-Artifact rejection is mentioned, but some details would be helpful. i.e. what was the criterion of rejection and how much data was excluded based on these steps?

I have a few comments regarding figures:

-As this is an experiment focusing on vision, a task figure showing the stimuli is necessary. A basic schematic of the stimuli and experimental design for both experiments should be shown.

-While spectrograms are show, it would be helpful to show one plot of the band limited time course for gamma ranges studied – providing a more accurate view of the amplitude response over time.

-Related to this, did the authors observe any color related effects in event related field potential? (prior work has reported a chromatic visual evoked potential).

-All figures should have both axis labels and units (e.g. color bars should have labels that provide the variable and its unit type).

Discussion:

-In the discussion, the authors correctly mention some differences in the magnitude of color responses across prior studies – highlighting the likely attenuation produced by more macroscopic measurements. However, given the striking similarities across other experimental tasks (e.g. that intracortical and surface recordings both show no gamma oscillations below ~10-15% grating contrast, suggesting similar measurement sensitivity), it seems that stimulus selection plays an equal role. As acknowledged by the authors elsewhere, it seems important to highlight how color selection, and in particular its saturation, may account for differences between studies/species.

6. PLOS authors have the option to publish the peer review history of their article (what does this mean?). If published, this will include your full peer review and any attached files.

Reviewer #1: No

Reviewer #2: No

---

## [Author Response · Author response to Decision Letter 0]

14 Oct 2020

Dear editor,

We would like to thank the reviewers for their helpful comments on the previous draft of our manuscript, “The gamma response to colour hue in humans: evidence from MEG”. In light of these comments we have now made several revisions to the manuscript. Our responses to the individual comments, and details of changes we have made to the manuscript to address them, are given below. We have also taken this opportunity to make some minor changes to the text for readability and have corrected a mistake in how we had plotted error bars in the previous draft of the manuscript. As part of the resubmission we include a version of the new draft containing mark-up showing all changes made to the text since the previous draft.

We hope these changes are sufficient for the manuscript to be published in Plos One but are happy to make further changes if you or the reviewers consider this necessary.

Best wishes,

Gavin Perry (on behalf of the authors)

Reviewer #1: The study of Perry et al is devoted to the MEG gamma responses induced by color hues. The recent invasive studies in animals and intracranial studies in humans have shown that homogeneous color surfaces induce gamma oscillations in visual cortex. This study is the first one that shows that the similar gamma responses can be recorded noninvasively in humans using MEG. The authors also investigated how the gamma response to the hue change is affected by chromatic separation between the colors (in perceptual color space). They confirmed the previously shown ‘preference’ for the red hue. They also described different gamma frequency components induced in MEG by hues.

This is an interesting and useful study. I think that the methods and results are clear. I only have minor comments and suggestions.

1. Please, describe what software has been used for the source localization analysis.

We have added the following (line 229): “Calculation of the forward model, covariance matrices and beamformer weights were performed using propriety software supplied by the MEG manufacturer (CTF). Subsequent analysis was performed in MATLAB using a combination of the Fieldtrip toolbox [36] and custom scripts.”

2. Line 199. What is the ‘approximate area of medial occipital cortex’? Please provide more information.

We’ve altered the line (now line 203) to try making it clearer how we selected the location of the virtual sensor: “The location of the local maximum within occipital cortex was found for each image (where multiple maxima were present with occipital cortex, the maximum closest to the occipital pole was selected).”

3. It might worth to show the ‘color separations’ used in the 2nd experiment in a figure similar to Fig.6.

This is already plotted in Figure 2. In order to draw the reader’s attention to this in the results section we have added the following text: “The coordinates of these stimuli in the CIELIUV colour space are shown in Fig 2.” (Line 343)

4. The ‘red’ color in the low separation condition is not really red, but rather orange. Could it be better to use the word ‘reddish’, ‘bluish’. …? 

For readability we prefer to stick the standard hue names, but to highlight this issue we have added the following to line 159: “Note that this meant that the colour hue used in the low separation conditions did not always matched the ‘named’ hue (e.g. in the low separation red-green pair, the ‘red’ stimulus had an orange hue, while in the red-blue pair the ‘red’ was closer to magenta).”

5. It took me some time to understand the design of the 2nd experiment. To the convenience of the lazy reader, I would suggest to insert a color line that illustrates the sequence of color stimuli above each panel of the Figure 7.

We have added these to the figure (now Figure 9) as suggested.

6. In the figure 9 I would suggest to change ‘Red’ to ‘Red-after-Blue’, ‘Red-after-Green’, etc. to improve readability of the figure.

We have amended the figure (now Figure 11) along the lines suggested.

7. Do you have an idea why the offset of the red hue promotes gamma response to the green and blue hues? This worths mentioning in the Discussion section.

We speculate that this may reflect the gamma response being primarily driven by the L-M opponent pathway, which were modulated strongly by the red-blue and red-green stimuli, but only weakly by the blue-green stimulus. We’ve added some text to the section of the discussion where we discuss the possible role of colour opponent mechanisms to make his more explicit: “This could explain why the response to the green and blue hues was stronger when preceded by a red hue, as this would have produced stronger temporal modulation of L-M opponent processes, whereas the green-blue pair would have produced very little L-M opponent response.” (Line 428) 

8. Figure 5.

A) Traces show one second of unfiltered data (centred on stimulus onset) from a single occipital sensor (MRO42) from each of the 35 trials of red visual stimulation from the first block of the experiment.

It is unclear what ‘from each of the 35 trials’ refers to. It looks like the left panel (A) shows only one epoch… 

B) Please provide units for the scale.

The previous version was confusing as it showed the same sensor over several epochs but was plotted in a format more normally used to show several sensors over the same epoch. We have changed the format of the figure to try to make it clearer that the data reflects 35 different epochs (i.e. sections from each of the 35 trials). We have also added scale information to the bottom right of panel A. 

9. Line 404. …. we found in Experiment 1 that the frequency of the gamma response was reduced for the green stimulus compared to the other hues,…

The frequency of the gamma peak for the green hue does not seem to be reduced (Fig. 3). The ‘low gamma’ for the green seems have even higher frequency than that for other hues.

We’ve added some text to make it clearer what we’re referring to here: “(in Fig 4 the mid frequency response to the green stimulus corresponds to the ‘bump’ around 47 Hz in the spectrum)” (Line 270)

Reviewer #2: Perry and colleagues utilized MEG measurements in human subjects to study the influence of stimulus color on visual gamma activity. In their first experiment, the authors show color effects consistent with recent monkey and human studies – i.e. gamma responses occur to uniform color stimuli, which were maximal red stimuli. In the second experiment, the authors employed three primary colors at two different chromatic distances to evaluate the impact of the preceding color presentation on the recorded response. The authors report that the chromatic separation between pairs of stimuli has an effect on the gamma response, although this effect is not equal across different color-pairs, showing a bias for pairs of colors involving red. This latter result is interesting and in line with evidence that the bias for red color can be partially related to adaptation effects although there is more to be addressed for understanding the origin of such a bias. Overall, the manuscript provides an important follow up to recent work and will be of interest to many. Below I highlight a few points of improvement related to the quantification and presentation of findings.

Major comments:

-Overall the quantification of observed effects could be improved. Results for experiment 1 seem to be mostly reported in Figure plots with modest statistical reporting. For example, what were the specific mean responses to each color? Saying post-hoc tests satisfied a corrected p-value is limited, provide the test statistic and related mean values.

We have added two new tables (tables 3 & 4) that give the mean amplitudes for each condition and the test statistics for pairwise comparisons respectively.

-Artifact rejection is mentioned, but some details would be helpful. i.e. what was the criterion of rejection and how much data was excluded based on these steps?

We have added the further detail on artefact rejection from line 185 on: “Artefact rejection was performed offline by manually inspecting the data and discarding epochs with large muscle or head-movement-related artefacts (because this process is inherently subjective, a single researcher performed artefact rejection in each experiment to avoid inter-rater differences in visual artefact identification). The mean number of rejected trials for each participant was 21 in Experiment 1 (maximum number of trials for any individual subject: 121) and 6 in Experiment 2 (maximum: 19).”

I have a few comments regarding figures:

-As this is an experiment focusing on vision, a task figure showing the stimuli is necessary. A basic schematic of the stimuli and experimental design for both experiments should be shown.

We have added a task figure as Figure 1.

-While spectrograms are show, it would be helpful to show one plot of the band limited time course for gamma ranges studied – providing a more accurate view of the amplitude response over time.

We agree that this provides some additional information in Experiment 1 and so we have added these plots as Figure 5. For Experiment 2 the plots of the timecourse mainly consists of the broadband onset/offset transients for each colour presentation which we think is already fairly clear from the spectrograms (Figure 9) and so we have not added a timecourse plot for that experiment. 

-Related to this, did the authors observe any color related effects in event related field potential? (prior work has reported a chromatic visual evoked potential).

In Experiment 1 we are able to measure event-related field to the colour stimuli. The tuning of the ERF to colour is similar to the tuning of mid frequency gamma (with the exception of the response to purple, which produces an ERF equal in amplitude to that of red). We performed a preliminary analysis of the ERFs in Experiment 2, but the data was frankly quite messy with not all the subjects showing a clear response. As the novel aspect of the research is the exploration of the gamma response to colour using MEG (there is already a pre-existing literature measuring the event-related response to colour in MEG/EEG) our preference is focus the manuscript on that data, rather than to present ERF data as well.

-All figures should have both axis labels and units (e.g. color bars should have labels that provide the variable and its unit type).

We have ensured that all figures now have axis labels and units.

Discussion:

-In the discussion, the authors correctly mention some differences in the magnitude of color responses across prior studies – highlighting the likely attenuation produced by more macroscopic measurements. However, given the striking similarities across other experimental tasks (e.g. that intracortical and surface recordings both show no gamma oscillations below ~10-15% grating contrast, suggesting similar measurement sensitivity), it seems that stimulus selection plays an equal role. As acknowledged by the authors elsewhere, it seems important to highlight how color selection, and in particular its saturation, may account for differences between studies/species.

We think it unlikely that colour selection explains the difference between studies as the red hue in Experiment 1 is quite close to the hue used by Shirhatti & Ray that produced such a strong response in macaques. We’ve added some text to this effect: “Firstly, this may reflect the fact that the colour hues used here did not exactly match those used by Shirhatti & Ray, due to the use of a different display device. However, we note that the red hue used in this study was relatively close to that used by Shirhatti & Ray (see Table 1 for comparison) and thus we might expect it to have produced a similarly strong response in this study.” (Line 452). The stimuli we used in Experiment 1 tended to be no closer to the white point in the colour space than the corresponding stimuli used by Shirhatti & Ray (see Table 1), meaning that they would have been at least as saturated in appearance as the stimuli used by S & R. Thus, the weaker responses (relative to the response to the grating) in our study cannot be explained by the stimuli being less saturated than those used in S & R’s study.

---

## [Decision Letter · Decision Letter 1]

18 Nov 2020

The gamma response to colour hue in humans: evidence from MEG

PONE-D-20-14792R1

Dear Dr. Perry,

I am pleased to inform you that your manuscript has now been judged scientifically suitable for publication and will be formally accepted for publication once it meets all outstanding technical requirements.

Thank you for choosing *PLOS ONE* for reporting your research. 

Kind regards,

Sasha 

Alexander N. "Sasha" Sokolov, Ph.D.

Academic Editor

PLOS ONE

Additional Editor Comments (optional):

Reviewers' comments:

Reviewer's Responses to Questions

**Comments to the Author**

1. If the authors have adequately addressed your comments raised in a previous round of review and you feel that this manuscript is now acceptable for publication, you may indicate that here to bypass the “Comments to the Author” section, enter your conflict of interest statement in the “Confidential to Editor” section, and submit your "Accept" recommendation.

Reviewer #1: All comments have been addressed

Reviewer #2: All comments have been addressed

2. Is the manuscript technically sound, and do the data support the conclusions?

Reviewer #1: Yes

Reviewer #2: Yes

3. Has the statistical analysis been performed appropriately and rigorously? 

Reviewer #1: Yes

Reviewer #2: Yes

4. Have the authors made all data underlying the findings in their manuscript fully available?

Reviewer #1: Yes

Reviewer #2: Yes

5. Is the manuscript presented in an intelligible fashion and written in standard English?

Reviewer #1: Yes

Reviewer #2: Yes

6. Review Comments to the Author

Reviewer #1: All my questions and comments were addressed. I recommend accepting the manuscript for publication.

Reviewer #2: The authors have provided a clear response to prior comments, which I believe will improve the clarity of results for future readers. These findings provide an important non-invasive replication of observations made in humans and non-human primates, making an important contribution to the literature. I have no further comments.

7. PLOS authors have the option to publish the peer review history of their article (what does this mean?). If published, this will include your full peer review and any attached files.

Reviewer #1: No

Reviewer #2: No

---

## [Editor Report · Acceptance letter]

7 Dec 2020

PONE-D-20-14792R1 

The gamma response to colour hue in humans: evidence from MEG 

Dear Dr. Perry:

I'm pleased to inform you that your manuscript has been deemed suitable for publication in PLOS ONE. Congratulations! Your manuscript is now with our production department. 

Kind regards, 

on behalf of

Dr. Alexander N. Sokolov 

Academic Editor

PLOS ONE